# Modified Over-the-Row Machine Harvesters to Improve Northern Highbush Blueberry Fresh Fruit Quality

**Lisa Wasko DeVetter [1,\*]**, **Wei Qiang Yang [2]**, **Fumiomi Takeda [3]**, **Scott Korthuis [4]** **and Changying Li [5]**

1   Northwestern Washington Research and Extension Center, Department of Horticulture, Washington State University, 16650 State Route 536, Mount Vernon, WA 98273, USA

2   North Willamette Research and Extension Center, Department of Horticulture, Oregon State University, 15210 NE Miley Road, Aurora, OR 97002, USA; wei.yang@oregonstate.edu

3   United States Department of Agriculture, 2217 Wiltshire Road, Appalachian Fruit Research Station, Kearneysville, WV 25430, USA; fumi.takeda@usda.gov

4   Oxbo International Corporation, 270 Birch Bay Lynden Road, Lynden, WA 98264, USA; skorthuis@oxbocorp.com

5   College of Engineering, University of Georgia, 200 D.W. Brooks Drive, Athens, GA 30602, USA; cyli@uga.edu

\*   Correspondence: lisa.devetter@wsu.edu; Tel.: +1-360-848-6124

**Abstract:** Improved blueberry mechanical harvesting (MH) equipment that maintains fresh market quality are needed due to rising costs and decreasing availability of laborers for harvesting by hand. In 2017, a modified over-the-row (OTR) blueberry harvester with experimental catch surfaces and plates designed to reduce fruit bruising was evaluated. The catch surfaces were made of neoprene (soft catch surface; SCS) or canvas (hard catch surface; HCS) and compared to hand-picked fruit (control). Early- and early/mid-season 'Duke' and 'Draper', respectively, were evaluated in Oregon, while late-season 'Elliott' and 'Aurora' were evaluated in Washington. Harvested berries were run through commercial packing lines with fresh pack out recorded and bruise incidence or fresh fruit quality evaluated during various lengths of cold storage. The fresh pack out for 'Duke' and 'Draper' were 83.5% and 73.2%, respectively, and no difference was noted between SCS and HCS. 'Duke' fruit firmness was highest among MH berries with SCS, but firmness decreased in storage after one week. Firmness was highest among hand harvested 'Draper' followed by MH with SCS. For 'Elliott' and 'Aurora', fruit firmness was the same across harvesting methods. 'Draper' exhibited more bruising than 'Duke', but bruise ratings and the incidence of bruising at ≤10% and ≤20% were similar between hand and MH 'Draper' with SCS after 24 h of harvest. 'Aurora' berries had similar bruise ratings after 24 h between hand harvesting and MH with SCS, while 'Elliott' showed more bruise damage by MH with both SCS and HCS than hand harvested fruit. Although our studies showed slightly lower fresh market blueberry pack outs, loss of firmness, and increased bruise damage in fruit harvested by the experimental MH system compared to hand harvested fruit, higher quality was achieved using SCS compared to HCS. We demonstrated that improved fresh market quality in northern highbush blueberry is achievable by using modified OTR harvesters with SCS and fruit removal by either hand-held pneumatic shakers or rotary drum shakers.

**Keywords:** harvest mechanization; hand-held shaking device; shaking equipment; fresh fruit quality

## 1. Introduction

Northern highbush blueberry (*Vaccinium corymbosum* L.) produces fragile and perishable berries that have been traditionally harvested by hand to maximize quality and postharvest longevity when sold on the fresh market. However, obtaining sufficient labor for harvest operations is increasingly difficult due to the rising costs of labor coupled with the decreasing availability of workers. Mechanizing and automating harvest operations is an important and growing area of research among blueberry and other specialty crop growers. The goal in developing viable machine harvest technologies for fresh market blueberry is to engineer equipment that can efficiently harvest berries while maintaining the quality and postharvest longevity necessary for this sector of the market.

Machine harvesting research for northern highbush blueberry began in the 1950s, but to date has met limited success for fresh market operations. Portable, hand-held shakers and over-the-row (OTR) machines that shake berries off bushes have been the primary areas of research focus. Hand-held shakers have been developed to harvest blueberries (Haven Harvesters, South Haven, MI, USA), but adoption of hand-held electric shakers has been limited because cultivars have variable fruit detachment rates that impact harvest efficiency [1,2]. Ergonomics of hand-held shakers is another concern, as the vibrations and range of motions may cause musculoskeletal strain on operators [3].

Recent technological advancements in light-weight pneumatic and electric hand-held shakers have improved harvest efficiency relative to hand harvest across several cultivars of highbush blueberry [2]. Furthermore, these shakers can be utilized with frames that have soft fruit catching surfaces, which can reduce bruise damage by lessening the impact forces of harvesting. Reduced bruise damage has been observed with use of soft fruit catching surfaces and research has shown percent bruise area in 'Draper' was 1.5% when picked with pneumatic hand-held olive shakers (Campagnola Inc., Bologna, Italy) with soft catching surfaces, whereas it was 3% when harvested by hand [2]. Similarly in southern highbush blueberry (complex hybrids of *V. corymbosum* and *V. darrowii* Camp; cvs. Chickadee, Flicker, and Kestrel), harvesting berries by hand-held pneumatic shakers resulted in 80% harvest efficiency and 90% pack-out with remaining berries being immature and overripe. These advancements in hand-held shakers highlight the need for continued research to assess their practicality for commercial blueberry harvesting.

While OTR machines have now become standard industry practice for machine harvesting blueberries for processed markets, only the V45 blueberry harvester (BEI Inc., South Haven, MI, USA) has harvested northern highbush blueberries with quality comparable to hand harvest [4]. Takeda et al. [5] showed that the V45 blueberry harvester has the potential to harvest southern highbush blueberry and rabbiteye blueberry (*V. virgatum* Aiton) with fruit quality approaching that of hand-harvested berries. Harvest efficiency was also improved by selective removal of vertically growing and overarching canes in the center of the bush. Commercial adoption has been limited, however, because of the need for a specific plant architecture for improved harvest efficiency and to limit plant damage, as well as the machine's low ground speed [5,6].

Recent surveys conducted in the United States and British Columbia, Canada, show blueberry growers are increasingly using OTR machines designed to pick berries for the processed market for their fresh market blueberry [7]. While diminished fruit quality is a concern, deciding on a harvest method (i.e., hand versus machine) is complex and impacted by a variety of factors that were documented in this survey work. Those factors include market price, availability and cost of labor, impacts on fruit quality, cultivar characteristics, and machine harvesting costs. Although not documented in this survey work, the anticipated time in postharvest storage will also influence harvest method with berries that will be rapidly sold on the fresh market being more suitable for machine harvesting than berries that will remain in storage for multiple weeks. This survey work also documented growers' concerns regarding increasing labor costs and their interest in new technologies that reduce these harvesting constraints.

A resurgence in machine harvesting research for fresh market blueberry occurred in 2008 with the funding of a large multi-disciplinary grant, "Advancing Blueberry Production Efficiency by Enabling Mechanical Harvest, Improving Fruit Quality and Safety, and Managing Emerging Diseases (United States Department of Agriculture National Institute of Food and Agriculture Specialty Crop Research Initiative program; Award No. 2008-51180-19579). Another large multi-disciplinary grant was funded in 2014 titled, "Scale Neutral Harvest Aid System and Sensor Technologies to Improve Harvest Efficiency and Handling of Fresh Market Highbush Blueberries" (United States Department of Agriculture National Institute of Food and Agriculture Specialty Crop Research Initiative program; Award No. 2014-51181-22383). Several machine harvesting systems have been evaluated through the latter project, including pneumatic hand-held shakers mounted on a portable catcher system with soft fruit catching surfaces and modified OTR harvesters with soft fruit catching surfaces and catcher plates installed inside the machine [8]. These prototypes were evaluated in 2016 and 2017 in Florida, California, Oregon, and Washington.

The aim of the 2017 work in Oregon and Washington was to advance harvest technologies for fresh market northern highbush blueberry using hand-held pneumatic shakers and OTR machine harvesters with rotary drum shakers combined with new soft catching surfaces designed to minimize berry bruising. Long-term, this research will contribute to the development of commercially available equipment and technologies that will enable machine harvesting of fresh market blueberry with high fruit quality and postharvest longevity.

## 2. Materials and Methods

### 2.1. Machine Harvesting Field Trials

Berries were harvested from northern highbush blueberry plants in Oregon and Washington during the 2017 growing season using a modified Oxbo 7420 harvester (Oxbo International Corp., Lynden, WA, USA). Modifications to the harvester included installation of experimental soft catch surfaces on a catch frame suspended above the catch plates and conveyor belts. This was done to reduce fruit dropping distance and bruising. In Oregon, a neoprene soft catch surface was used and the plastic catch plates (e.g., 'scales' or 'fish plates') were hollowed out and a neoprene material was also installed on the top side of each catch plate (Figure 1a). This treatment was abbreviated SCS (soft catch surface) for 'soft catch system'. On the other side of the harvester, a canvas catch surface was installed over the conveyor belt with standard plastic catch plates. This treatment was abbreviated HCS (hard catch surface) for 'hard catch system'. Long-handled, pneumatically-operated, hand-held olive harvesters (Campagnola, Bologna, Italy) removed berries from the bush and were operated by workers standing on a platform inside the harvester. There were two workers on either side of the machine harvesting fruit. The machine was modified in Washington whereby the canvas catch surface was replaced with neoprene so that both sides of the machine had neoprene catch surfaces (Figure 1b). The canvas catch surface was replaced with neoprene because preliminary data indicated that the neoprene surface lead to less bruising incidence in harvested fruit. The catch plates and treatment abbreviations remained the same. Furthermore, mechanical Orbirotor® picking heads (i.e., rotary drum shakers; Oxbo International Corp., Lynden, WA, USA) were installed and used instead of pneumatic shakers. Catching surfaces, either SCS or HCS, were considered our experimental treatments and compared to hand harvested fruit (control).

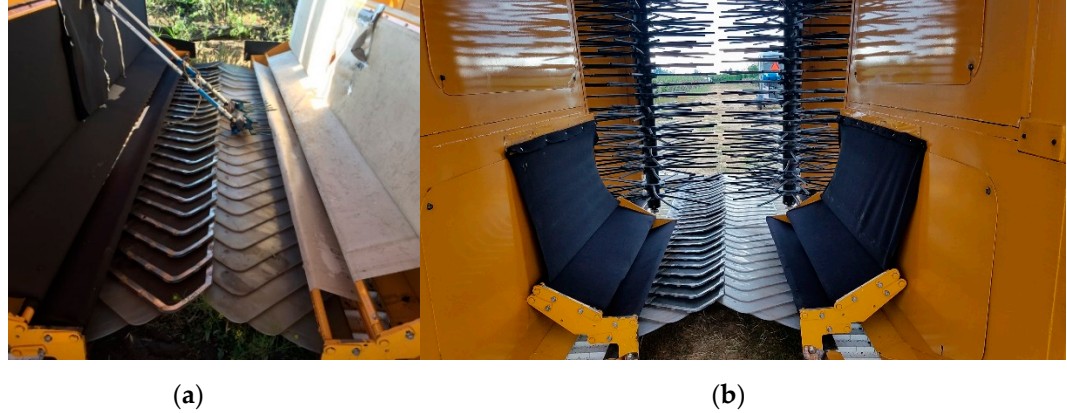

(**a**)                                                                    (**b**)

**Figure 1.** Modifications to an Oxbo 7420 machine harvester for fresh market blueberry. The harvester in Oregon had a neoprene soft catch surface suspended above the conveyor belt and hollowed out catch plates with neoprene installed on the top side of each catch plate (left; **a**). A canvas catch surface was installed over the conveyor belt with standard plastic catch plates on the other side of the machine (right; **a**). In Washington, both sides of the machine had neoprene catch surfaces, but the catch plates were the same as in Oregon (**b**). Hand held pneumatic shakers were used to harvest 'Duke' and 'Draper' blueberries in Oregon. 'Elliott' and 'Aurora' blueberries were harvested with Orbirotor® picking heads (i.e., rotary drum shakers) in Washington.

Berries from early-season 'Duke' and early/mid-season 'Draper' were harvested in commercial fields in Salem, Oregon (lat: 45° N, 122° W). Berries were hand-harvested once and then machine harvested 10 days later. Machine harvesting with the experimental OTR harvester occurred 11 July 2017 for 'Duke' and 31 July for 'Draper'. Rows of 'Duke' and 'Draper' measuring 152 m and 146 m, respectively, were used for the study. Each row was evenly divided into four sections that were 38 m long for 'Duke' and 36.5 m long 'Draper'. Sections were treated as replicates in a completely randomized design. As the machine passed over the row, fruit were harvested using the SCS or HCS. Two conveyer belts on top of the machine conveyed fruit to lugs for filling and there was one conveyer belt per treatment. The conveyer belts were cleared between replicates. The machine was operated so that approximately 10 lugs filled half-full and weighing 4.5–6.8 kg each were collected per replicate. Hand harvested fruit were collected from an adjacent row. After machine and hand harvest, berries were taken to a commercial packing plant located on site. Field heat was removed by forced air-cooling to an internal fruit temperature of 10 °C. Pre-cooling temperature conditions were approximately 4–7 °C. Pre-cooled berries were packed into 170 g covered plastic clamshells after two and four days for 'Duke' and 'Draper', respectively, through standard packing lines. The fresh pack-out including percent of blue fruits, colored fruits, debris, and soft fruits were calculated using sort-outs retrieved from the packing line and weighted. Replicates were treated as separate lots on the packing line in order to avoid mixing treatments and replicates.

Late-season 'Elliott' and 'Aurora' berries were evaluated on a commercial berry farm in Lynden, Washington (lat: 48.9° N, 122.6° W). Both hand and machine harvesting occurred the morning of 31 August 2017. The farm provided a trained machine harvest operator and hand-picking crew for this experiment. Single cultivar rows were divided into four sections that were 18 m long each; each section was treated as a replicate in a completely randomized design. Similar to Oregon, fruit were harvested using SCS or HCS (no canvas; hard plastic plates only) with two conveyer belts on top of the machine conveying fruit to lugs. There was one conveyer belt per treatment and conveyer belts were cleared between replicates. The machine was operated so that 10 lugs filled half-full and weighing 4.5–6.8 kg each were collected per replicate. Two lugs per replicate were randomly collected and transported to a nearby packing facility in Sumas, Washington for sorting and packing. Berries from an adjacent row of 'Elliott' and 'Aurora' were simultaneously hand-picked into lugs and two lugs per cultivar were also transported to the packing facility. No pack-out data were collected at this facility. Berries were

first precooled as described in Oregon before being packed into 454 g covered plastic clamshells on a refrigerated packing line. Again, replicates were treated as separate lots on the packing line in order to avoid mixing treatments and replicates.

## 2.2. Fruit Quality Assessments

Packed fruit were stored in a refrigerated room at 1 °C for one day until they were picked up and transported to university labs for quality analyses. Berries were stored in refrigerated rooms at 1 °C for two and four weeks in Washington and Oregon, respectively. Firmness and bruising evaluations occurred 24 h after harvest and after two weeks of storage. In Oregon, fruit firmness was also evaluated after one, three, and four weeks of cold storage. All fruit quality evaluations were conducted 4–6 h after removal of the fruit from cold storage to allow the fruits to warm to room temperature. Firmness was measured from 50 berries in Oregon and 25 berries in Washington per replicate using a FirmTech II (Bioworks, FirmTech II, Bioworks, Wamego, KS, USA). The FirmTech had maximum and minimum compression forces of 250 g and 25 g, respectively, in Oregon. In Washington, the maximum compression force was 200 g and the minimum compression force was 15 g. Incidence of bruising was visually assessed from 25 or 50 berries per replicate (50 berries in Oregon and 25 berries in Washington) by measuring the level of pulp discoloration due to senescence and bruising. To assess percent bruise area, berries were cut perpendicular to the fruit axis and assigned a rating based on the severity of pulp darkening due to water soaking and/or pigment bleeding [9]. Ratings were based on a scale ranging from 0 to 100. A rating of 0 indicates no bruising, while a rating of 100 indicates berries showed black water-soaking across the entire cut surface. The number of berries with ≤10% and ≤20% bruise area were also determined. Previous studies indicated that fruit with less than 25% of the sliced surface area showing bruise damage could be held in cold storage for several weeks or more and still retain fresh market quality [10,11].

## 2.3. Statistical Analysis

Data were assessed to determine if they met the criteria for Analysis of Variance (ANOVA). Pack out data were analyzed as a one-way ANOVA with cultivar differences compared by paired *t*-test in SAS (Statistical Analysis System software, Ver. 9.3, SAS Institute, Inc., Cary, NC, USA). Firmness and bruise were analyzed using SAS GLM. Means separation by the least-square means procedure at $p \leq 0.05$ was done with Bonferroni adjustments.

## 3. Results

### 3.1. Pack-Out of 'Duke' and 'Draper'

The percent of packed blue fruit that were marketable differed by cultivar, but were similar between MH (mechanical harvesting) with HCS and SCS (Table 1). Average blue pack out for 'Duke' was 83.5% and 73.2% for 'Draper'. Color defects were greater for 'Draper' regardless of catch surfaces and were due to green and/or red berries. 'Draper' also had a greater percentage of trash and hand sort outs. However, 'Duke' had more defects due to softness relative to 'Draper' and softness was slightly elevated with the HCS.

**Table 1.** Pack out of 'Duke' and 'Draper' blueberries after machine harvesting with hand-held pneumatic shakers used by workers standing on a platform built into a modified over-the-row harvester prototype with experimental hard and soft catch surfaces (HCS and SCS, respectively) in Oregon in 2017

| | | Defects (%) | | | |
|---|---|---|---|---|---|
| **Treatment** | **Blue Pack out (%)** | **Color** | **Trash** | **Hand Sort outs** | **Soft** |
| Duke | | | | | |
| HCS | 84.0 | 7.7 | 1.4 | 1.2 | 5.8 |
| SCS | 83.1 | 9.1 | 1.6 | 1.4 | 4.7 |
| Draper | | | | | |
| HCS | 72.7 | 18.3 | 3.0 | 3.0 | 3.1 |
| SCS | 73.7 | 17.4 | 2.7 | 2.9 | 3.4 |
| Significance [1] | | | | | |
| Duke vs. Draper | <0.0001 | <0.0001 | <0.0001 | <0.0001 | <0.0001 |
| HCS vs. SCS | NS | NS | NS | NS | NS |

[1] Cultivar means were compared using a paired *t*-test $\alpha = 0.05$, while differences between catch surfaces were determined by one-way ANOVA (Analysis of Variance) for each cultivar. NS denotes not statistically significant.

*3.2. Firmness*

'Duke' and 'Draper' fruit firmness was more responsive to MH treatment when compared to 'Elliott' and 'Aurora'. Firmness was greatest for 'Duke' harvested using the SCS at 24 h and one week postharvest (Table 2). At two weeks, 'Duke' fruit firmness harvested with SCS was the same as hand and lowest among fruit harvested using HCS. Machine picked 'Duke' continued to lose firmness at a faster rate than hand-picked berries. By three weeks postharvest, berries harvested with SCS and HCS had lower firmness than hand-picked 'Duke', although berries harvested with SCS were firmer than those harvested with HCS. 'Draper' fruit firmness was greatest when harvested by hand throughout the study. Among machine harvested 'Draper', berries were firmer when harvested using SCS relative to HCS after one to three weeks of cold storage. Harvest method had no effect on fruit firmness of 'Elliott' and 'Aurora' 24 h and two weeks after cold storage (Table 3).

**Table 2.** Firmness (g/mm) of 'Duke' and 'Draper' blueberries 24 h and 1, 2, 3, and 4 weeks after machine harvesting with hand-held pneumatic shakers used by workers standing on a platform built into a modified over-the-row harvester prototype with experimental hard and soft catch surfaces (HCS and SCS, respectively) in Oregon. Machine harvested berries were compared to hand-harvested (control) fruit in 2017.

| Treatment | 'Duke' | | | | 'Draper' | | | | |
|---|---|---|---|---|---|---|---|---|---|
| | **24 h** | **1 week** | **2 weeks** | **3 weeks** | **24 h** | **1 week** | **2 weeks** | **3 weeks** | **4 weeks** |
| HCS | 174 b [1] | 160 b | 150 | 123 c | 167 b | 166 c | 137 c | 133 c | 127 c |
| SCS | 184 a | 170 a | 169 | 135 b | 171 b | 175 b | 149 b | 144 b | 141 b |
| Hand (control) | 156 c | 158 b | 161 | 143 a | 185 a | 190 a | 175 a | 176 a | 166 a |
| *p*-value | 0.0001 | 0.0017 | NS | 0.0001 | 0.018 | 0.0001 | 0.0004 | 0.0001 | 0.019 |

[1] Means followed by the same lower case letter within a column are not statistically different at $\alpha = 0.05$; NS denotes not statistically significant.

**Table 3.** Firmness (g/mm) of 'Elliott' and 'Aurora' blueberries 24 h and 2 weeks after machine harvesting using a modified over-the-row harvester prototype with Orbirotor® picking heads and experimental hard and soft catch surfaces (HCS and SCS, respectively) in Washington. Machine harvested berries were compared to hand-harvested (control) fruit in 2017.

| Treatment | 'Elliott' | | 'Aurora' | |
|---|---|---|---|---|
| | 24 h | 2 weeks | 24 h | 2 weeks |
| HCS | 145 [1] | 121 | 164 | 161 |
| SCS | 155 | 123 | 172 | 159 |
| Hand (control) | 160 | 109 | 157 | 178 |
| *p*-value | NS [2] | NS | NS | NS |

[1] Twenty-five berries per one of four replicates were evaluated at each sampling time. [2] NS denotes means within a column are not different at α = 0.05.

## 3.3. Bruise Area

Bruise ratings determined 24 h after harvest were lowest in hand-harvested 'Duke' and similar between hand-harvested and MH 'Draper' (Table 4). Bruise incidence at ≤10% was highest in hand-harvested 'Duke' 24 h after harvest and were the same across treatments at <20%. No differences were detected in 'Duke' bruise ratings two weeks after harvest. 'Draper' exhibited more bruising than 'Duke'. However, bruise ratings and the incidence of bruising at ≤10% was similar between hand and MH 'Draper' after 24 h of harvest. After two weeks, both MH 'Draper' with SCS and HCS had a higher bruise rating than hand-harvested berries and the incidence of bruising was also less among hand harvested berries.

**Table 4.** Incidence of bruising in 'Duke' and 'Draper' blueberries 24 h and 2 weeks after machine harvesting using a modified over-the-row harvester prototype with experimental hard and soft catch surfaces (HCS and SCS, respectively) and pneumatic shakers in Oregon. Machine harvested berries were compared to hand-harvested (control) fruit in Oregon in 2017.

| Treatment | 24 h after Harvest | | | 2 weeks after Harvest | | |
|---|---|---|---|---|---|---|
| | Bruise Rating [1] | ≤10% Bruise | ≤20% Bruise | Bruise Rating | ≤10% Bruise | ≤20% Bruise |
| Duke | | | | | | |
| HCS | 6 a [2] | 89 b | 95 | 19 | 47 | 73 |
| SCS | 4 b | 93 b | 97 | 16 | 57 | 83 |
| Hand (control) | 2 c | 99 a | 100 | 19 | 53 | 79 |
| *p*-value | <0.0001 | 0.0159 | NS [3] | NS | NS | NS |
| Draper | | | | | | |
| HCS | 16 a | 45 b | 77 | 24 a | 39 b | 67 b |
| SCS | 12 b | 62 a | 89 | 22 a | 41 b | 65 b |
| Hand (control) | 14 ab | 57 ab | 81 | 12 b | 70 a | 87 a |
| *p*-value | 0.0366 | 0.0463 | NS | <0.0001 | 0.0001 | 0.0045 |

[1] Ratings were on a 0 to 100 scale with 0 indicating no bruising and 100 indicating the entire cut surface was bruised. [2] Fifty berries per one of four replicates were evaluated at each sampling time; Means followed by the same lower case letter within a column are not statistically different at α = 0.05. [3] NS denotes not statistically significant.

'Elliott' had greater overall internal bruise damage compared to 'Aurora' (Table 5). Bruise ratings determined 24 h after harvest were greatest for fruit MH with HCS for both 'Elliott' and 'Aurora'. 'Elliott' hand-harvested berries had the lowest bruise rating 24 h after harvest, whereas 'Aurora' hand-harvested berries had the same bruise rating as MH with SCS. No differences in bruise ratings were observed two weeks postharvest for 'Elliott', whereas 'Aurora' blueberries MH with HCS and SCS had greater bruise ratings compared to hand harvested berries. The ≤10% and ≤20% bruise evaluations also showed less internal bruising for berries harvested by hand, although berries MH with SCS tended to have less internal bruising than MH with HCS 24 h after harvest. Percentage of

bruised berries did not differ for 'Elliott' two weeks after harvest. Machine harvested 'Aurora' berries had the same incidence of bruising two weeks after harvest regardless of catch surface and was greater than hand-harvested berries.

**Table 5.** Incidence of bruising in 'Elliott' and 'Aurora' blueberries 24 h and 2 weeks after machine harvesting using a modified over-the-row harvester prototype with Orbirotor® picking heads and experimental hard and soft catch surfaces (HCS and SCS, respectively) in Washington. Machine harvested berries were compared to hand-harvested (control) fruit in Washington in 2017.

| Treatment | 24 h after Harvest | | | 2 weeks after Harvest | | |
|---|---|---|---|---|---|---|
| | Bruise Rating [1] | ≤10% Bruise | ≤20% Bruise | Bruise Rating | ≤10% Bruise | ≤20% Bruise |
| Elliott | | | | | | |
| HCS | 36 a [2] | 17 b | 24 c | 59 | 14 | 22 |
| SCS | 29 b | 17 b | 42 b | 54 | 17 | 22 |
| Hand (control) | 18 c | 42 a | 67 a | 43 | 18 | 35 |
| *p*-value | <0.0001 | 0.0172 | 0.0034 | NS [3] | NS | NS |
| Aurora | | | | | | |
| HCS | 16 a | 42 b | 69 b | 23 a | 40 b | 58 b |
| SCS | 11 b | 69 a | 81 ab | 22 a | 25 b | 61 b |
| Hand (control) | 10 b | 74 a | 89 a | 12 b | 68 a | 85 a |
| *p*-value | 0.0002 | 0.0283 | 0.0425 | <0.0001 | 0.0030 | 0.0056 |

[1] Ratings were on a 0 to 100 scale with 0 indicating no bruising and 100 indicating the entire cut surface was bruised. [2] Twenty-five berries per one of four replicates were evaluated at each sampling time; Means followed by the same lower case letter within a column are not statistically different at $\alpha = 0.05$. [3] NS denotes not statistically significant.

## 4. Discussion

Machine harvesting blueberries for fresh market is achievable and was demonstrated by high pack-out and fruit firmness that was similar to hand-harvested berries (Tables 1–3). Internal bruising increased with machine harvesting but was lessened with SCS (Tables 4 and 5). 'Draper' bruised more readily than 'Duke'. Bruising became more apparent two weeks postharvest in 'Aurora', while 'Duke' and 'Elliott' showed no differences among harvesting method after two weeks of storage. These observations demonstrate MH with SCS can be comparable to hand harvesting under certain situations. However, increased internal bruising could decrease shelf-life and/or fruit quality when placed under longer-term cold storage and modified atmosphere conditions may be needed to preserve the quality and integrity of the berries.

Mechanizing and automating harvest operations for fresh market blueberry is an important endeavor due to high labor costs and decreasing labor, which is challenging the viability of farming operations. Previous research has demonstrated harvest efficiency, labor productivity, and labor costs can be improved through mechanized harvesting [1,10,12]. Yet, the quality and postharvest longevity needed for fresh markets can be jeopardized [10,12–14]. The problem of quality and postharvest longevity is particularly problematic for northern highbush and southern highbush blueberries. Cultivars within these species can also respond differently to machine harvesting technologies, which underscores the need to assess machine harvesting across multiple species and cultivars of blueberry [1,5,15].

Differential responses of cultivar to harvesting method was observed in our study. For example, 'Draper' had less pack-out than 'Duke' and overall more defects that had to be sorted out on the packing line (Table 1). 'Draper' is a firm-fruited cultivar well-suited to machine harvesting for the processed market [16], but a high proportion of red and green fruits lead to reduced pack-out (Table 1). While 'Duke' pack-out was commercially acceptable, machine harvested 'Draper' will need to have improved pack-out at ≥80% before it is acceptable. Delaying harvest time to allow more berries to color could improve pack-out, but fruit firmness may be reduced due to overripe berries. Engineering

and/or operating the machines to have greater selectivity is another avenue to reduce harvest defects and improve pack-out of 'Draper' and similar cultivars.

Another cultivar effect was observed in 'Elliott'. Berries harvested from 'Elliott' bushes were overall very soft and had a high incidence of bruising and discoloration irrespective of harvesting method (Tables 3 and 5). This softness is likely why few differences were observed for this cultivar. As a late season cultivar in the northwest, 'Elliott' is known to produce soft berries compared to other northern highbush blueberry cultivars such as 'Duke' and 'Draper'. Adjusting harvest time to improve fruit firmness in 'Elliott' may not be ideal because the high acidity levels in its fruit decrease fruit quality. 'Elliott' therefore may not suitable for machine harvesting for the fresh market.

Fruit quality of 'Duke', 'Draper', and 'Aurora' in terms of firmness and internal bruising was the same or similar for hand harvested and MH fruit using SCS (Tables 2–5). Firmness and bruising tended to increase when MH with HCS. These findings support continued research and engineering using SCS over HCS. Reducing the drop height in OTR machines with SCS should be investigated to determine whether bruising incidence can be further reduced and fruit quality equivalent to hand harvest can be achieved across commercially important cultivars. New SCS will also need to be further tested for durability in order to determine whether they can withstand standard commercial harvesting operations. Equally important is that new SCS need to be tested for food safety, including biofilm formation and effectiveness of sanitization treatments.

Using semi-mechanical harvest aid systems like the OTR machine with hand-held shakers evaluated in this study improves harvest efficiency relative to hand harvest [2]. However, the system with hand-held shakers requires two to four operators to stand inside the machine to harvest the fruit. Removing the hand-held shakers and installing the Orbirotor® shaking heads improved the overall efficiency of the OTR system by eliminating the need to have two to four workers inside the machine operating the shakers. With increasing labor constraints, systems that reduce the need for workers are economically advantageous.

## 5. Conclusions

This work demonstrated machine harvesting blueberries with pack-out and fruit firmness comparable to hand-harvest is achievable using OTR harvesters combined with SCS. These SCS reduce fruit bruising by dampening the impact force of berries dropping into the harvester and can be integrated into conventional OTR harvesters with picking heads (i.e., rotary drum shakers). Further evaluations testing the efficiency, selectivity, and impacts on fruit quality and pack-out among other commercially important blueberry cultivars will be important to determine the viability of these modified harvesters for fresh market operations.

**Author Contributions:** Conceptualization, methodology, and investigation, F.T., W.Q.Y., C.L., S.K., and L.W.D.; Formal analysis and data curation, F.T., W.Q.Y., and L.W.D.; Resources, F.T., C.L., S.K., W.Q.Y., and L.W.D.; Writing—original draft preparation, L.W.D.; Writing—review and editing, F.T., W.Q.Y., C.L., S.K., and L.W.D.; Project administration and funding acquisition, C.L. and F.T.

**Funding:** This research was funded by U.S. Department of Agriculture, National Institute of Food and Agriculture, Specialty Crop Research Initiative research program, award no.: 2014-51181-22383.

**Acknowledgments:** We would like to acknowledge grower cooperators including Munger Farms, Pan-American Berry Growers, and Allen Brown at Maberry Packing, LLC. Technical assistance in the field and laboratory was provided by Brian Foote, Sean Watkinson, Weixin Gan, Huan Zhang, James Jedediah Smith, and Ann Rose.

**Conflicts of Interest:** The authors declare no conflict of interest. The funders had no role in the design of the study; in the collection, analyses, or interpretation of data; in the writing of the manuscript, or in the decision to publish the results.

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
