# Peer review of "Modified Over-the-Row Machine Harvesters to Improve Northern Highbush Blueberry Fresh Fruit Quality"

_agriculture, doi:10.3390/agriculture9010013_

Reviewer 1 Report

The manuscript “Modified Over the row machine harvesters to improve northern highbush blueberry fresh fruit quality”

is an interesting topic, however I think that the article needs of some major revisions. My suggestions and specific comments are included below.

It’s necessary to check the English language because some there are some errors.see line 26( use of plural,see line 172)

ABSTRACT: please check the lenght , i think it is too long.

MATERIAL & METHODS

Line 118  No plants but fruits were harvested The author define the different way of harvesting as treatments, but technically it is no possible define them treatments , soplease avoid the use of treatments in the text  and in the material and methods section. Please specify that fruit harvested by hand is the control sample , it’s no clear.

Line 146 No plants but fruits

Line 152  Pre- cooled fruits were processed? ( what authors mean?). It’s no clear) Whic type of clamshells were used? Materials? Are covered? Whic temperature of precoolinf was used?

Line 167 Whic lab?

For how many time fruit were stored?

Line 172-174 : Rewrite this sentence it is no clear.

WHy two weeks?

Why 25 or 50? In function of what?

Please delete tratments term from tables

Line 205 check points

Author Response

Dear Reviewer 1:

My co-authors and I would like to express our gratitude for the consideration of our article.  Please find enclosed a revised version of the manuscript, “Modified Over-the-Row Machine Harvesters to Improve Northern Highbush Blueberry Fresh Fruit Quality” (Manuscript ID: agriculture-414026).  We hope the revisions are sufficient and look forward to contributing to the scholarly work published in the journal.  

We are appreciative of the feedback.  Reviewer 1 comments have been addressed in this response letter by listing each comment separately and stating how we changed the manuscript accordingly in red font.  Changes have also been made in the attached manuscript and are highlighted in yellow. In situations where we felt a change was not needed, we have provided an explanation supporting our decision.  

§  The manuscript “Modified Over the row machine harvesters to improve northern highbush blueberry fresh fruit quality” is an interesting topic, however I think that the article needs of some major revisions. My suggestions and specific comments are included below. We are glad the topic is found to be interesting and appreciate your suggestions and comments.

§  It’s necessary to check the English language because some there are some errors. See line 26 (use of plural,see line 172). English is the first language of the lead author and the remaining authors also have English language proficiency. The manuscript has been re-reviewed for these types of issues should all be corrected.  

§  ABSTRACT: please check the lenght , i think it is too long. The revised abstract has been abbreviated to less than 350 words.

§  MATERIAL & METHODS

§  Line 118  No plants but fruits were harvested. We have changed to “Berries were harvested from northern highbush blueberry plants..,”

§  The author define the different way of harvesting as treatments, but technically it is no possible define them treatments , so please avoid the use of treatments in the text  and in the material and methods section. Please specify that fruit harvested by hand is the control sample , it’s no clear. We are unclear why the harvesting treatments cannot be defined as treatments and respectfully disagree. Our treatments were: 1) Harvesting fruit by hand (control); 2) Machine harvesting with neoprene soft surfaces designed to reduce bruising when the berries fall from the bush to the machine (called “soft catch system; SCS); and 3) Machine harvesting with canvas surfaces that similarly were designed to reduce bruising (called “hard catch system”; HCS). These treatments are fundamental to our experiment and overall project goal, which is to design machine harvesters that reduce bruising and lead to harvested fruit with quality comparable to hand-harvested fruit.  We have clarified that hand-picked fruit is our control.

§  Line 146 No plants but fruits. Changed to “Berries were…”

§  Line 152  Pre- cooled fruits were processed? ( what authors mean?). Pre-cooling is a very standard practice of removing field-heat through forced air-cooling. The preceding line in the manuscript explains this as, “Field heat was removed by forced air-cooling to an internal fruit temperature of 10 °C.” After pre-cooling, berries are placed on the packing line and packed into plastic clamshells. We have removed the word “processed” and just indicated the berries were packed, as we believe the word may be adding to the reviewer’s confusion.

§  It’s no clear) Which type of clamshells were used? Materials? Are covered? We have clarified they were covered plastic clamshells.

§  Which temperature of precooling was used? We have added this temperature range (4-7 °C).

§  Line 167 Which lab? We have specified “university labs”, as the work was conducted in the labs of both DeVetter and Yang.

§  For how many time fruit were stored?  Fruit were stored for 2 weeks in Washington and 4 weeks in Oregon. We believe the modification to lines 172-174 (below) clarifies this.

§  Line 172-174 : Rewrite this sentence it is no clear. These lines have been rewritten.

§  Why two weeks? We had developed a protocol to measure these variables after two weeks in Washington. In Oregon, it was decided to double the time to four weeks out of experimental interest, but this was not immediately communicated to the Washington location which is why the storage times differ between locations.

§  Why 25 or 50? In function of what?  The response to this is similar as above. We had developed a protocol and it was to measure 25 berries per replicate, as previous work has shown that to be a sufficient sample size. However, in Oregon, it was decided to double the sample size per replicate and this was not immediately communicated to the Washington location.

§  Please delete treatments term from tables. We respectfully do not feel we can delete the treatment terms from the tables, as deleting the treatment terms would make the reader unable to discern our treatments. We created tables with the goal that they are stand-alone and can be understood independent of the text.

§  Line 205 check points. We were not able to address this, as we do not understand this comment. Apologies!

Thank you for your comments and feedback,

Lisa & co-authors

Reviewer 2 Report

Abstract is too long, please check the length and describe the merit and results of the research.

May be poor literature for that article, there are more authors and articles related to this topic.

In Materials and Methods are some not so clear description of methodology, please explain more details of work. Please check terminology.

For better understanding of modification of machine, would be nice to put to the article technological scheme of working apparatus.

Author Response

Dear Reviewer 2:

My co-authors and I would like to express our gratitude for the consideration of our article.  Please find enclosed a revised version of the manuscript, “Modified Over-the-Row Machine Harvesters to Improve Northern Highbush Blueberry Fresh Fruit Quality” (Manuscript ID: agriculture-414026).  We hope the revisions are sufficient and look forward to contributing to the scholarly work published in the journal.  

We are appreciative of the feedback.  Reviewers' comments have been addressed in this response letter by listing each comment separately and stating how we changed the manuscript accordingly in red font.  Changes have also been made in the attached manuscript and are highlighted in yellow. In situations where we felt a change was not needed, we have provided an explanation supporting our decision.  

§  Abstract is too long, please check the length and describe the merit and results of the research. We have shorted the abstract to less than 350 words. Thank you.

§  May be poor literature for that article, there are more authors and articles related to this topic. Thank you for the concern. We included key papers that are relevant to this manuscript and area of research while balancing concerns regarding manuscript length. Furthermore, this team has been researching machine harvesting for fresh market blueberry for multiple years and we included key papers we are aware of and/or have contributed to that relate to this study. We do not believe adding additional articles would add substantially to this manuscript.  

§  In Materials and Methods are some not so clear description of methodology, please explain more details of work. Please check terminology. We have added more description to our methods based on another reviewer’s comments and re-checked the terminology. Please clarify if anything remains unclear.

§  For better understanding of modification of machine, would be nice to put to the article technological scheme of working apparatus. We appreciate and understand this request, but are unable to do so because of patenting concerns. We hope the photos of the machine and modifications included as figures are adequate.

Thank you, 

Lisa and co-authors

Round  2

Reviewer 1 Report

I agreed with the final manuscript  . It follow  the suggestion.